# Hypothesis-generating analysis of the impact of non-damaging metabolic acidosis on the transcriptome of different cell types: Integrated stress response (ISR) modulation as general transcriptomic reaction to non-respiratory acidic stress?

Virginie Dubourg[1☯], Marie-Christin Schulz[1☯], Philipp Terpe[1☯], Stefanie Ruhs[2], Michael Kopf[1], Michael Gekle[1]*

1 Julius-Bernstein-Institute of Physiology, Martin Luther University Halle-Wittenberg, Halle, Germany,
2 Klinik für Anästhesiologie und Intensivmedizin, Martin Luther University Halle-Wittenberg, Halle, Germany

☯ These authors contributed equally to this work.
* michael.gekle@medizin.uni-halle.de

## Abstract

Extracellular pH is an important parameter influencing cell function and fate. Microenvironmental acidosis accompanies different pathological situations, including inflammation, hypoxia and ischemia. Research focussed mainly on acidification of the tumour micromilieu and the possible consequences on proliferation, migration and drug resistance. Much less is known regarding the impact of microenvironmental acidosis on the transcriptome of non-tumour cells, which are exposed to local acidosis during inflammation, hypoxia, ischemia or metabolic derailment. In the present hypothesis-generating study, we investigated the transcriptional impact of extracellular acidosis on five non-tumour cell types of human and rat origin, combining RNA-Sequencing and extensive bioinformatics analyses. For this purpose, cell type-dependent acidosis resiliences and acidosis-induced transcriptional changes within these resilience ranges were determined, using 56 biological samples. The RNA-Sequencing results were used for dual differential-expression analysis (DESeq and edgeR) and, after appropriate homology mapping, Gene Ontology enrichment analysis (g:Profiler), Ingenuity Pathway Analysis (IPA®), as well as functional enrichment analysis for predicted upstream regulators, were performed. Extracellular acidosis led to substantial, yet different, quantitative transcriptional alterations in all five cell types. Our results identify the regulator of the transcriptional activity NCOA5 as the only general acidosis-responsive gene. Although we observed a species- and cell type-dominated response regarding gene expression regulation, Gene Ontology enrichment analysis and upstream regulator analysis predicted a general acidosis response pattern. Indeed, they suggested the regulation of four general acidosis-responsive cellular networks, which comprised the integrated stress response (ISR), TGF-β signalling, NFE2L2 and TP53. Future studies will have to extend the

**Data Availability Statement:** The analyzed data supporting the conclusions of this article are included within this article and its additional files. Additionally, raw RNA sequencing data are publicly available in the Gene Expression Omnibus (GEO) database (https://www.ncbi.nlm.nih.gov/geo). GEO accession number: GSE220788 (human data), GSE220789 (rat data).

**Funding:** This project was supported funded by the Deutsche Forschungsgemeinschaft (DFG GE 905/19). The funders had no role in study design, data collection and analysis, decision to publish, or preparation of the manuscript.

**Competing interests:** The authors have declared that no competing interests exist.

results of our bioinformatics analyses to cell biological and cell physiological validation experiments, in order to test the refined working hypothesis here.

## Introduction

Under physiological conditions, arterial pH is maintained within a narrow range (between pH 7.35 and 7.45). As for the interstitial fluid (i.e. the cellular microenvironment), its physiological pH-values range between pH 7.0 and 7.4, depending on the type of tissue and the actual physical condition (rest or exercise) [1–3]. The interstitial pH-value is unstable compared with the value in arterial blood and it may deviate from physiological values even when the arterial blood pH is unaffected [4]. Systemic metabolic acidosis, defined as a reduced serum bicarbonate [$HCO_3^-$] concentration in a person with physiological respiratory function, reflects a decrease in non-volatile acid removal that leads to net acid accumulation and subsequently to a general pH decrease. Prolonged systemic metabolic acidosis is accompanied by several consequences, including bone demineralization, skeletal muscle protein catabolism, reduced hepatic albumin synthesis, and increased systemic inflammation that results in an enhanced risk of cardiovascular and renal diseases [3].

Non-volatile acids derive from metabolism of endogenous or exogenous components, like amino acids, phospholipids or nucleic acids. In addition, glycolysis, ketone body synthesis (and incomplete oxidation fatty acids) contribute to acid accumulation [3]. The daily load of non-volatile acids in healthy adults on a Western style diet amounts to approximately 70 mmol/d, but can increase significantly in the event of pathological lipolysis or glycolysis (fasting?). Individuals with normal kidney and liver functions quantitatively match acid excretion to production, avoiding acid accumulation. In contrast, impaired kidney function can lead to acid accumulation, resulting in systemic metabolic acidosis even without any enhancement of acid production.

Locally, extracellular acidification with pH-values below 7.0 (in extreme situations down to 6.0) [3, 5–12] is one of the pathological hallmarks of many diseases. These include cerebral and cardiac ischaemia, cancer, infection, inflammation or tissue hypoxia. The development of local metabolic acidosis is the result of an imbalance in the production/release of acids by the cells and the removal of acids via the circulation. Extracellular acidosis acts on all cells of the affected tissue and may therefore lead to diverse cellular responses, resulting in the activation or aggravation of distinct pathophysiological processes.

In the past, research focussed mainly on tumour microenvironment acidosis as key parameter for cell fate and disease progression. However, prolonged acidification of the micromilieu also leads to inflammatory and fibrotic homeostatic imbalance in non-tumour tissues, characterized by the abnormal accumulation of immune cells, the appearance of myofibroblasts and interstitial fibrosis [3], from what a vicious cycle may result. Up to now, the molecular mechanisms of acidosis-associated inflammation and fibrosis are only partially understood. However, they seem to include both innate and adaptive inflammatory processes, as well as an imbalanced matrix homeostasis, resulting from dysregulated intra- and intercellular communications. Local acidosis can induce oxidative stress that can stimulate further inflammation and fibrosis, extending the damage to tissue function failure. In renal tissue for example, acidosis can induce an upregulation of angiotensin II, accompanied by leukocyte infiltration, proliferation and activation of fibroblasts and abnormal accumulation of extracellular matrix. In vascular walls, pH can drop significantly e.g. during severe sepsis but also—to a lesser extend–in atherosclerotic lesions, exposing smooth muscles to acidic conditions [10].

In addition, local acidosis can interfere with the function and differentiation state of paren-chymal cells, in what transcriptomic changes, most probably, play a major role [13]. Indeed, extracellular acidosis can be sensed at the cell membrane, what almost certainly leads to a decrease of cytosolic pH [14–16], thereby acting on a number of cellular processes, including signaling pathways, metabolic processes but also transcription regulation [3]. Recently, we described the impact of metabolic acidosis on cell signaling pathways and the expression of inflammation-related genes in fibroblasts and tumour cells [14, 15]. The results showed that an acidic microenvironment can trigger a differential transcriptional program of pathological relevant genes, generating an inflammatory status that leads to cellular loss of function. How-ever, it remains unclear whether there is a common global transcriptional "acidosis response" or whether individual responses for each cell type prevail.

In the present study, we performed a comparative investigation regarding the impact of non-damaging acidotic pH-values on the transcriptome of five non-cancerous cell types, com-prising epithelial and mesenchymal cells of human and rat origin. Because these cell-types are confronted with different acidotic challenges in their original environment, it is conceivable that they developed and exhibit different acidosis resiliences, i.e. the acidic pH-range that will not result in non-specific cell damage leading to cell death by necrosis or apoptosis. Therefore, it would not have been appropriate to expose the cells to a uniform acidic pH-value. Rather the cells had to be exposed to their individually determined maximum non-damaging acidotic pH values. Thus, we first assessed pH-resilience and subsequently exposed the cells to their respec-tive non-damaging pH-values for 48 h, followed by RNA-sequencing and bioinformatics anal-ysis (Fig 1).

## Materials and methods

### Cell culture

HAoSMC (male, caucasian) were cultivated in PromoCell Smooth Muscle Cell Growth Medium 2 containing 5% FCS (Fetal Calf Serum), epidermal growth factor (0.5 ng/ml), basic fibroblast growth factor (2 ng/ml) and insulin (5 μg/ml) at 37 ˚C in a humidified atmosphere with 5% $CO_2$. Normal rat kidney fibroblasts (NRK-49F, ATCC[®] CRL-1570), normal rat kid-ney epithelial cells (NRK-52E, ATCC[®] CRL-1571), normal human kidney epithelial cells (HK-2, CRL-2190™) and normal human fibroblasts (CCD1092Sk, ATCC[®] CRL-2114™) were grown in DMEM medium supplemented with 10% FCS and 2 g/l $NaHCO_3$ at 37 ˚C and under a humidified 5% $CO_2$ atmosphere, and subcultivated once per week before confluence.

Before all experiments, cells were synchronized by incubation in serum- and supplement-free DMEM media (5.5 mM glucose, 24 mM NaHCO3, 25 mM HEPES) for 24 h. Subse-quently, cells were treated in DMEM medium under following conditions for 48 h: control pH 7.4 or acidic pH as indicated. Acidic pH-values were obtained by titration with hydrochloric acid.

### Caspase-3 activity assay

Protein fractions of all cell types were obtained after 30 min incubation on ice with 100 μL cell lysis buffer (10 mM TRIS, 100 mM NaCl, 1 mM EDTA, 0.01% triton X-100 (v/v), pH 7.5) and used to determine caspase-3 activity. 60 μl of sample were incubated with 60 μl of Caspase-reaction buffer (20 mM piperazine-N,N′-bis(2-ethanesulfonic acid), 4 mM EDTA, 0.2% 3-[(3-cholamidopropyl)dimethylammonio]-1-propanesulfonate (w/v), 10 mM dithiothreitol, pH 7.4) and 42 μM DEVD-AFC (end concentration) for 90 min at 37 ˚C. Fluorescence of the cleaved product AFC was measured with a plate reader (Infinite M200, Tecan) at a 400 nm excitation and 505 nm emission wavelengths. Cleaved AFC was quantified using a calibration

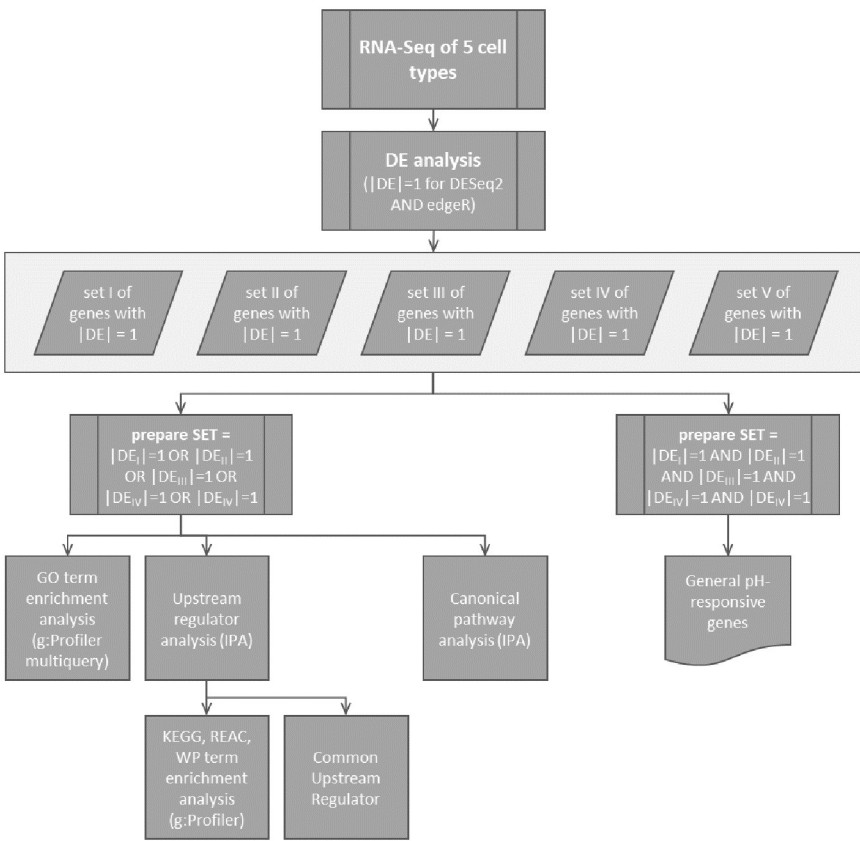

**Fig 1. Analysis workflow.** Analysis strategy for comparisons and identification of the acidosis-responsive genes, regulators and pathways, based on the output of the differentially expression (DE) analysis, which served to identify up- (DE = 1) and down- (DE = -1) regulated genes.

curve with known AFC concentrations and normalized to the total protein amount of the sample (determined by BCA assay, as described below).

## Lactate dehydrogenase (LDH) assay and cellular protein

LDH was determined after a standard protocol according to Bergmeyer et al. [17]. Cell media and cell lysates were incubated with LDH substrate buffer and the turnover of LDH substrates was measured at 334 nm (NADH) for 30 minutes. Relative LDH release was calculated as the LDH activity in the media divided by the total LDH activity (= media + lysate). Protein content was determined using a bicinchoninic acid assay (BCA—Thermo Scientific) with bovine serum albumin as standard [18].

## Determination of cytosolic pH and transporters

Cytosolic pH of single cells was determined using the pH-sensitive dye BCECF (2',7'-bis-(2-carboxyethyl)-5-(and-6)-carboxyfluorescein, acetoxymethyl ester—Invitrogen, Paisley, UK) as described before [19, 20]. In brief, cells were incubated with media containing 5 µM BCEC-F-AM for 15 min. Excitation light source was a 100 W mercury lamp. The excitation wavelengths were 450 nm/490 nm. The emitted light was filtered through a bandpass-filter (515–565 nm). After background subtraction, fluorescence intensity ratios were calculated. pH

calibration was performed after each experiment by the nigericin (Sigma, St. Louis, USA) technique [21, 22], using a two-point calibration (pH 6.8 and 7.5). The calibration solutions contained 132 mM KCl and 1 mM $CaCl_2$, 1 mM $MgCl_2$, 10 mM HEPES and 10 μM nigericin.

## RNA sample preparation

Total RNA were isolated after 48h treatment with BlueZol Reagent as described in the user manual. The RNA samples were treated with "Turbo DNAse-free kit" (following the "rigorous DNAse treatment" protocol from the manufacturer) to remove eventual genomic DNA contaminations and were cleaned by ethanol precipitation (with 3M sodium acetate, glycogen and 100% ethanol). The RNA concentration was determined by NanoDrop (Biochrom, Germany). The quality of the to-be-sequenced RNA samples was assessed using a 2100 Bioanalyzer system (Agilent Technologies, Germany) and all samples had a RNA Integrity Number (RIN) above 7 (with 10 as maximal possible value).

## RNA-sequencing

Novogene Co., Ltd (Cambridge, United-Kingdom) carried out the sequencing libraries preparation (poly(A) enrichment) and the paired-end sequencing (2 x 150 bp) runs on a Nova-Seq6000 Illumina system. Adaptor clipping and data quality control was provided by the service company as well.

HISAT2 (v. 2.1.0) [23] served for read mapping to the human genome hg38 for human cells (HK2, CCD1092Sk and HAoSMC) and to the rat genome rn7 for rat cells (NRK-52E, NRK-49F). Counting of the mapped reads was performed with featureCounts 2.0 (-p–M–t exon) [24] and gene annotation was done using BiomaRt (v.2.44.4) [25] to access Ensembl archive v105. Raw RNA sequencing data and annotated counts are publicly available on Gene Expression Omnibus (GEO) database (https://www.ncbi.nlm.nih.gov/geo). GEO accession numbers: Human data GSE220788, rat data GSE220789.

## Differential expression analysis

edgeR (3.30.3) [26] and DESeq2 (1.28.1) [27] were used to identified acidosis-regulated genes in each cell-types (Fig 1). Because not all cell types were from a single organism type, the differential expression analysis was divided in two parts, considering either human or rat cells. For each analysis round, genes with sufficient counts to be considered in the statistical analysis were filtered using the filterByExpr edgeR function and the independent filtering parameter (α = 0.05) of the DESeq2 results function. Normalization factors were calculated with the "trimmed mean of M value" (TMM) method in the edgeR analysis. Significantly "differentially expressed genes" (DEG) were defined as genes with a false discover rate (FDR) below 0.05 in both DESeq2 and edgeR outputs (overlap of the respective results).

## Homology mapping

The function getLDS from the BiomaRt (v 2.44.4) R package [25] was used for homology mapping between the human and rat genome annotations (Ensembl 105). Although the homology system is originally asymmetrical, a strategy to obtain an index with 1 human gene ID:1 rat gene ID had to be developed in order to be able to compare the acidosis-regulated genes in between all cell types (Fig 2).

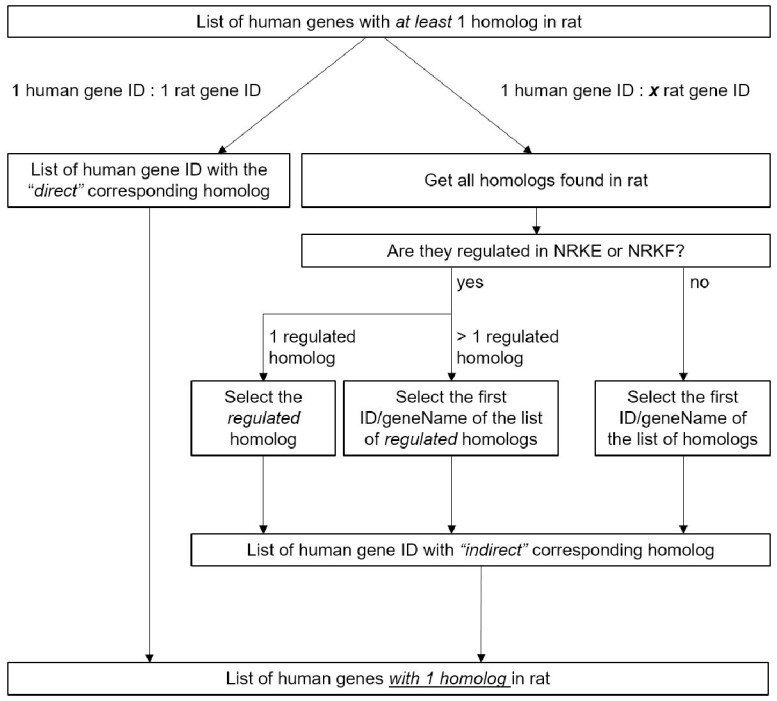

**Fig 2. Analysis workflow.** Strategy applied in order to obtain a symmetrical homology system between human and rat annotations used for further comparisons of the acidosis-regulated genes.

## Comparison of gene expression regulation and visualization

UpSetR (1.4.0) [28] was used to display the results of the comparison of these homolog-annotated lists of regulated genes. Clustering of the samples was performed with the MORPHEUS tool (https://software.broadinstitute.org/morpheus; metric: one minus pearson correlation, linkage: average; accessed on 16 February, 2023). For clustering according to the FPM values, a FPM-threshold > 5 for at least one sample was applied, resulting in 16126 genes to be included. For clustering according to the logFC values, a threshold of |logFC| > 1 for at least one cell type was applied, resulting in 6743 genes to be included.

## Gene Ontology enrichment

Gene Ontology (GO) term enrichment analysis was performed with the web server g:profiler2 (https://biit.cs.ut.ee/gprofiler/orth) [29]. A multiple query was performed to compare all cell types. For each considered cell type, a list of acidosis-regulated genes with at least one homolog in the other organism served as input. Only GO terms comprising less than 5000 genes were considered and those were simultaneously filtered for an adjusted p-value below 0.05 for all datasets. The enrichment score E of the filtered GO terms was calculated, with E = (intersection size/query size) / (term size/effective domain size).

## Ingenuity Pathway Analysis

The application QIAGEN Ingenuity Pathway Analysis (IPA—https://digitalinsights.qiagen.com/IPA) was used to compare the putative effect of acidosis-induced transcriptomic changes in the different cells types. The Ensembl identifiers of the regulated genes were mapped to

networks incorporated into the software database. The featured "Comparison Analysis" tool was used to match the different results of the "Upstream Regulator" (predicts potential regulators involved in the observed gene expression variations—includes e.g., transcription or translation regulator, growth factor, kinase, chemicals) and the "Canonical Pathways" (predicts potentially affected pathways downstream of the observed gene expression variations) analyses. Only the category "Genes, RNAs and Proteins" was considered for "Upstream Regulators" (exclude e.g., described chemicals). The latter were further filtered for "Transcriptions Regulators" before heatmap visualization. All results were filtered for $|Z\text{-score}| \geq 2$ and adjusted (Benjamini–Hochberg) p-value $\leq 0.01$ for at least one dataset.

### Functional enrichment analysis for predicted upstream regulators

The functional enrichment analysis of predicted upstream regulators (output of IPA analysis, filtered for adjusted p-value $\leq 0.05$ for all datasets) was performed with the web server g:profiler2 (https://biit.cs.ut.ee/gprofiler/orth) [29]. Only biological pathways-related databases were considered (KEGG, Reactome, WikiPathways). The results were filtered for adjusted p-values $\leq 0.001$ and enrichment score $E > 10$ (see "Gene Ontology Enrichment" section for E calculation method).

## Results

### Determination of cellular acidosis resilience

Because the cell types used in this study are confronted with different acidotic challenges in their original environment, they most probably developed different acidosis resilience, i.e. are able to cope with different acidic pH-ranges that will not result in non-specific cell damage leading to cell death by necrosis or apoptosis. As it would not have been appropriate to expose the cells to a uniform acidic pH-value rather than to expose them to their individual maximum non-damaging acidotic pH values, we first determined cell type-specific acidosis resilience.

Fig 3 shows the effect of exposure to acidic media on caspase-3-activity (biomarker for apotosis), release of cytosolic proteins (LDH or caspase-3; biomarker for necrosis, i.e. membrane leakage) and cell protein per petri dish (biomarker for cell loss) for the different cell types. Obviously, the cells differ in acidosis resilience. As already described before, NRK-52E and NRK-49F cells show a high acidosis resilience down to pH 6.0 [15]. HK-2 and CCD1092Sk cells showed signs of cell damage (necrosis) beyond pH 6.4 and HAoSMC beyond pH 6.8. Thus, NRK-52E and NRK-49F cells were exposed to pH 6.0, HK-2 and CCD1092Sk cells to pH 6.4 and HAoSMC to pH 6.8.

### Acidic-milieu triggers gene expression regulation with organism- and cell type-specific patterns

The incubation with acidic media led to gene expression changes in all considered cells types (Fig 4a, left; see also the summaries in the S1 File and S1 Fig), with a visible higher disposition for rat cells (S1 Table). The higher number of affected genes in NRK-49F and NRK-52E cells could in part result from the lower extracellular pH ($pH_e$). However, this seems unlikely (Fig 4g) considering the fact that (i) there was a rather large difference in the number of regulated genes between these two rat cells types exposed to the same acidic stress and (ii) in view of the higher number of affected genes in HAoSMC (pH 6.8) compared to HK-2 or CCD1092Sk cells (pH 6.4). Furthermore, there seems to be no correlation of intracellular pH values with the number of genes affected (Fig 4g). We assume that the differences result from a predominant cell-type driven rather than a pH-value-driven transcriptional response.

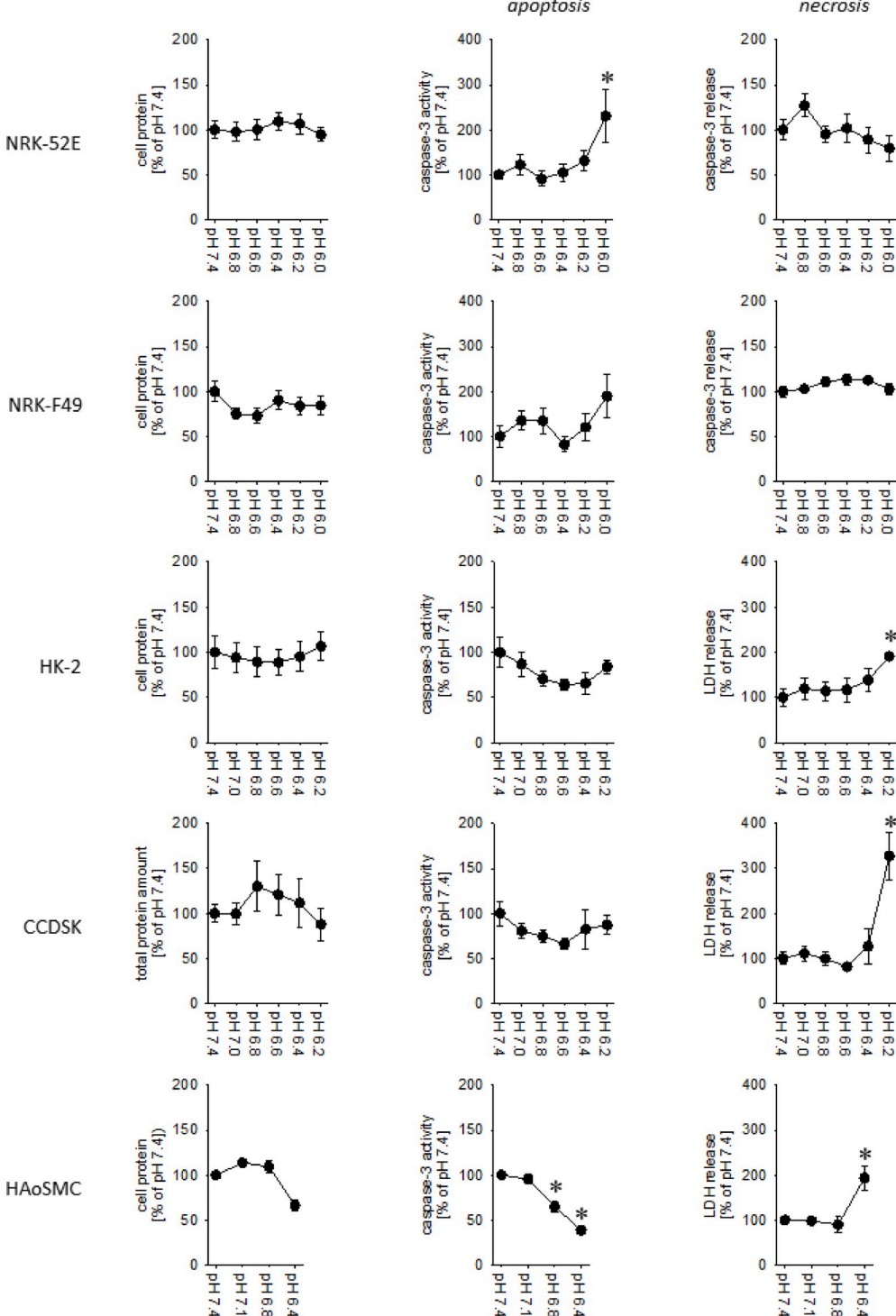

**Fig 3. Determination of acidosis resilience for the five cell types.** Cell protein per dish (left), caspase-3-activity as apoptosis marker (middle) and relative release of cytosolic proteins (LDH or caspase-3; right) as necrosis marker. N = 6–10. * = p<0.05 versus pH = 7.4 (Mann-Whitney Rank Sum Test).

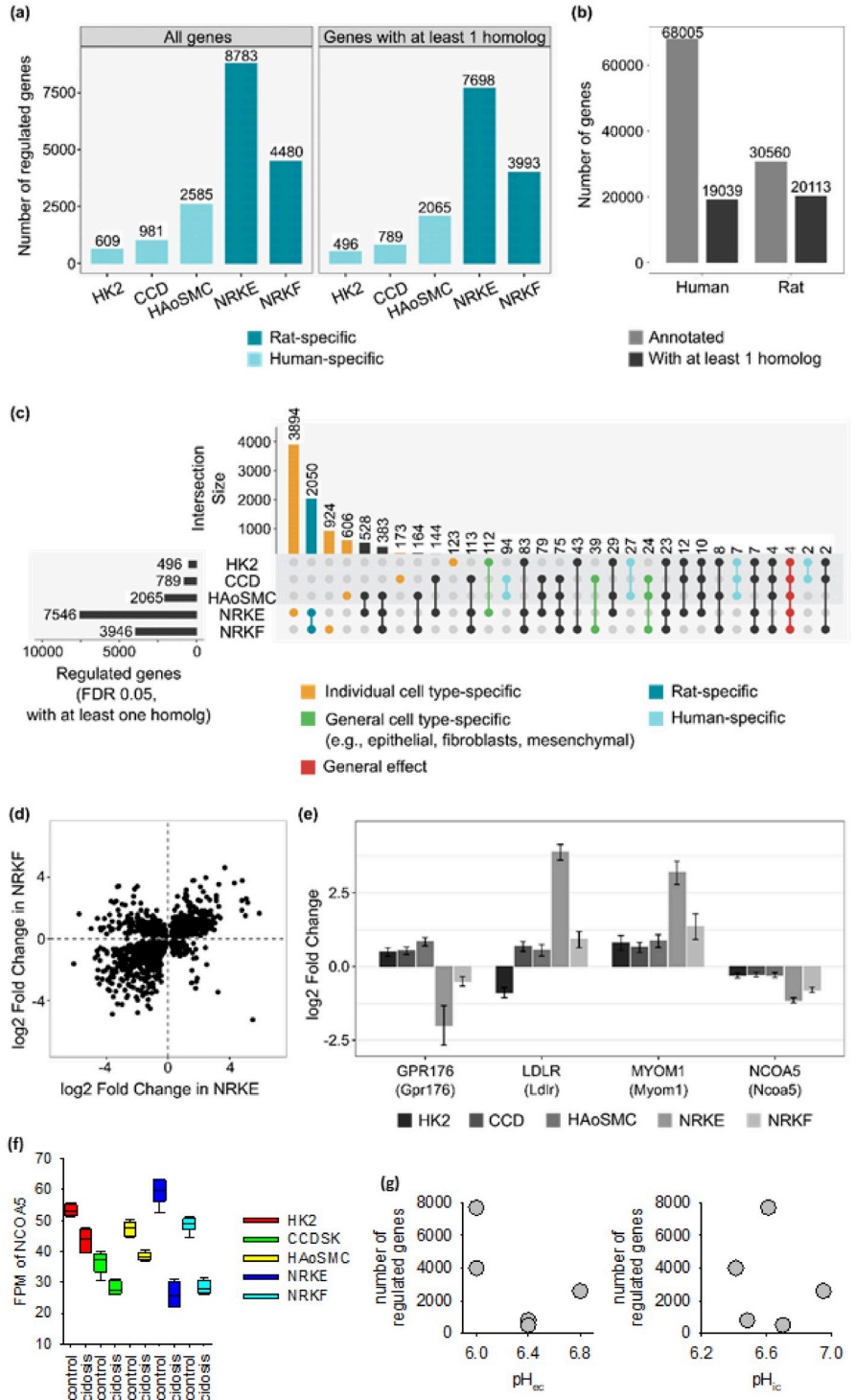

**Fig 4. Acidic milieu mostly induces organism- and cell type-specific gene expression regulations.** (a) Numbers of regulated genes (FDR 0.05) per cell type before and after filtering for genes having at least one homolog in the other considered organism. (b) Numbers of annotated genes for each genome and the proportion of these genes having at least one homolog in the other considered organism, according to Ensembl homology annotation. (c) The overlaps of the different lists of regulated genes are displayed in an UpSet plot, in which each row corresponds to a set of regulated genes and each column to one segment of a hypothetical Venn diagram. A black/coloured or grey dot indicates that genes from the corresponding dataset are included or not in this intersection, respectively. (d) The visualization of the log2 fold changes of the genes regulated in both rat cell types (second intersection of the UpSet plot) shows that not all

of them are regulated in the same direction, highlighting a cell type specificity. (e) log2 fold changes for the genes found regulated through all datasets annotation ("general effect" intersection of the UpSet plot). The error bars corresponds to the log2 fold change standard error calculated by DESeq2. (f) Expression of NCOA5 under control and acidosis conditions in the five cell types. (g) Number of regulated genes does not correlate with the degree of intra- or extracellular acidosis (values for HK2 and CCD were derived from the literature [30–33]).

All genes annotated for the human and rat genomes were filtered for genes having at least one homolog in the other species (Fig 4b). These lists of homolog-associated genes were then used to screen the acidosis-regulated genes. This induced a slight reduction of the considered numbers of regulated genes only (Fig 4a, right), showing that most of the acidosis-regulated genes have at least one homolog in the other considered species. The homology system between human and rat is nonetheless asymmetrical as more than one rat gene ID can be associated with more than one human gene ID, and vice versa. Aiming to compare the acidosis-regulated genes throughout all the considered cell types, we developed a strategy to overcome this asymmetrical distribution of the homolog genes (Fig 2) and obtained a 1 human gene ID:1 rat gene ID homology. Based to this approach, the lists of acidosis-regulated genes in each cell type were compared, indifferently to which the organism the data were linked (Fig 4c). This suggested that acidosis partly induced organism-specific gene expression patterns ("rat- or human-specific" intersections). Cell type specific regulation patterns were also observed, either for genes regulated exclusively in one cell type ("cell type specific" intersections) or for genes whose regulation direction depended on the cell type (as an example Fig 4d shows the comparison of NRK-52E and NRK-49F cells). Clustering of all samples according to their FPM values (see Methods section for details) also highlighted species- and cell type-specificities (Fig 5a). Cells clustered first according to the species, then according to the cell type (including embryological origin—epithelial versus mesenchymal for human cells) and only then according to acidotic stress. Clustering of all samples according to their logFC values (see Methods section for details) also highlighted species- and cell type- as well as cell origin- specificities (Fig 5b and 5c). Finally, four genes were regulated throughout all datasets ("general effect" intersection in Fig 4c), highlighting a reduced but yet substantial global effect of acidosis on gene expression. Out of these, two genes were regulated concordantly in all five cell types, MYOM1 and NCOA5. As the abundance of myocyte-specific MYOM1 was very low (< 3 FPM) in four cell types, the general biological relevance of this acidosis response is most probably neglectable. Whether it is relevant for myocytes cannot be decided from our data. By contrast, the abundance of NCOA5, an ubiquitous transcription co-regulator, was substantial in all five cell types, indicating that the expression of this gene represents a general pH-sensor mechanism (Fig 4f).

### Functional enrichment analysis by g:Profiler

We compared the five data sets for common enriched GO terms (S2 Table). Only 21 GO terms passed our significance thresholds for all datasets. The enrichment score of these GO terms was below 2.5, with the highest mean enrichment for GO term GO:0045859 (regulation of protein kinase activity). Thus, there seem to be no directlyacidosis-related GO terms strongly enriched by transcriptional regulation. Nevertheless, closer inspection of these 21 GO terms revealed that they refer to cellular phosphorus metabolism, protein phosphorylation and cytoskeleton organisation, biological processes known to be pH-sensitive.

### Comparative upstream analysis of the transcriptional changes

Fig 6a gives an overview of the "Upstream Regulator Analysis" output after filtering for predicted upstream regulators annotated as "transcription regulators". The heatmap format gives

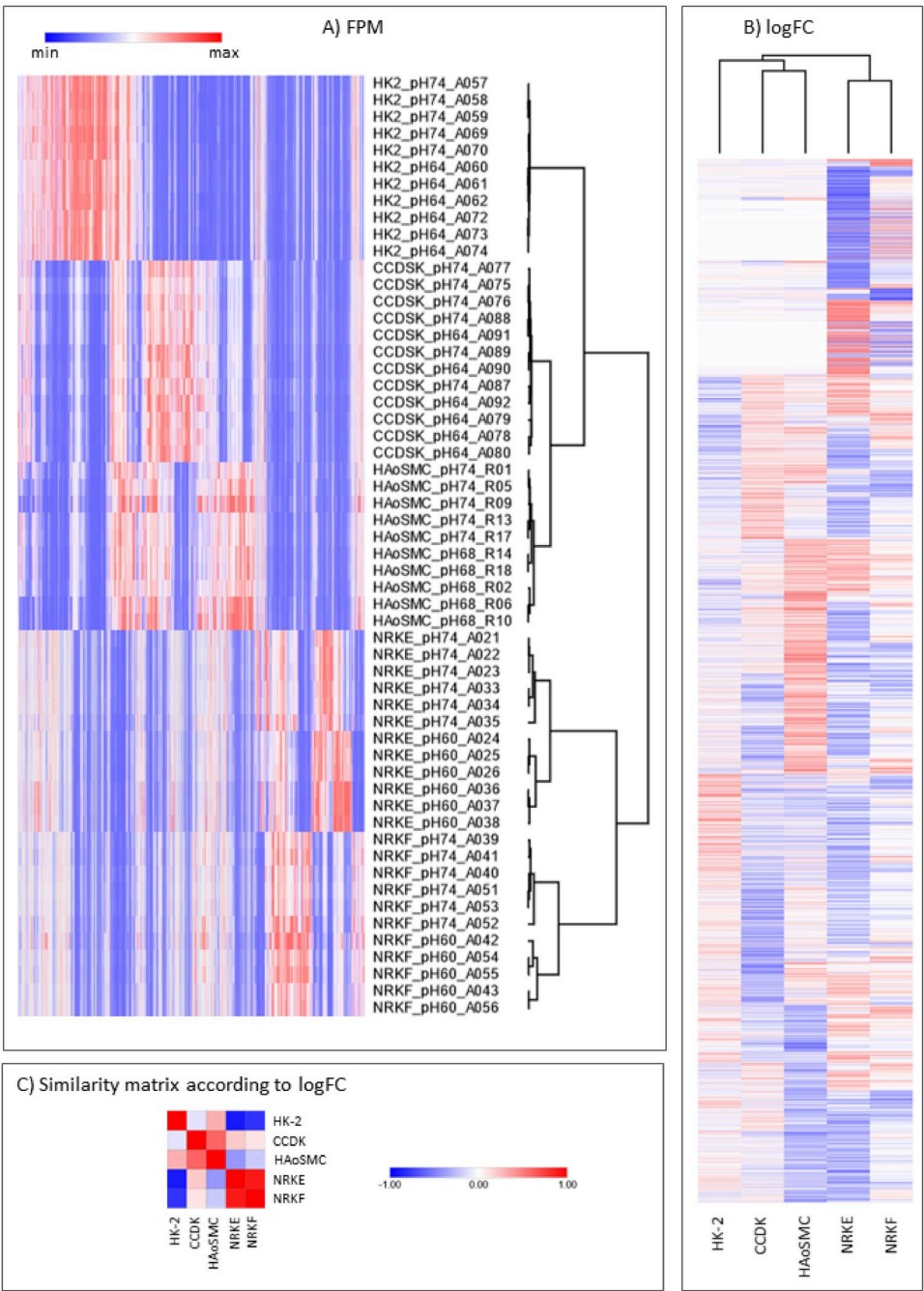

**Fig 5. Acidosis induces organism- and cell type-specific gene expression patterns.** The heatmaps show the FPM (a) and the normalized log2 fold changes (b, c) for all genes regulated in at least one cell type, with each row corresponding to one gene. The column clustering resulted in a species- and cell-specific clustering.

the opportunity to visualize eventual regulation patterns throughout the datasets. It thus showed that no transcription regulator was predicted to be regulated identically in all cell types. But it also highlighted that they may be activated (or inhibited) in a species-specific manner, as several clusters of transcription regulators were specifically predicted for rat datasets. The hierarchical clustering supported this observation as human and rat datasets clustered

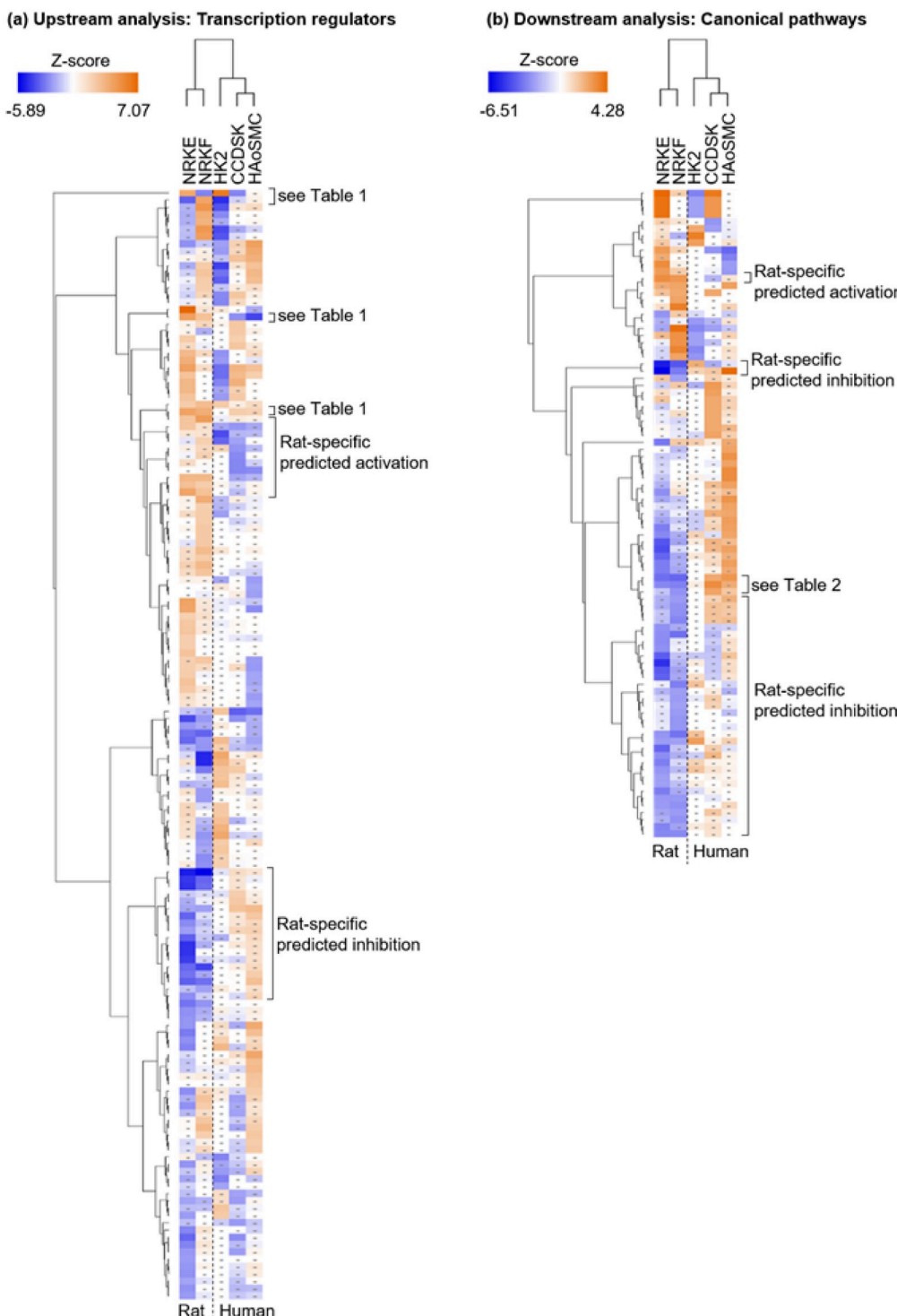

**Fig 6. Upstream and downstream analyses for acidosis-induced gene expression regulation.** The present heatmaps (generated by IPA) aim to give an overview of the results (detailed values are available in S3 Table). Each row corresponds to a transcriptional regulator (a) or to a canonical pathway (b) predicted as significantly activated/inhibited for at least one dataset. The colour scales are related to the calculated Z-scores (with a positive Z-score corresponding to a putative activation and a negative Z-score to a putative inhibition). A dot indicates that an upstream regulator or a canonical pathway did not reach the threshold |Z-score| ≥ 2 for the corresponding dataset. The hierarchical clustering used Euclidean distance metric.

separately. Additionally, the heatmap in Fig 6a featured upstream regulators predicted specifically for epithelial cell or fibroblasts datasets, independently of the species, therefore highlighting a putative cell type-specific gene expression regulation.

Table 1 illustrates these different patterns by showing detailed values for four selected predicted upstream regulators, which all displayed high absolute Z-scores (all detailed values associated with Fig 6a are available in S3 Table). ATF4, a transcription factor, was predicted as activated for rat datasets but as inhibited for two out of three human datasets, showing therefore a species-specific pattern of potential regulation. On the other hand, the transcriptional coactivator NUPR1 displayed a cell type-specific regulation, with a putative activation in epithelial cells but an inhibition in fibroblasts. Conversely, the transcription factor CEBPB is predicted to be inhibited in epithelial cells and activated in (rat) fibroblasts. Finally, the transcription factor NFE2L2, which has been suggested to play a role cell response to acidosis [13], is the only one putatively activated throughout four out of five the cell types. Thus, NFE2L2 may represent a more general principle regarding cellular responses to acidosis. NUPR1, ATF4, NFE2L2 and CEBPB are well known transcriptional regulators that act in response to cell stress, mainly oxidative and endoplasmic reticulum stress, and regulate cell survival [34–38]. Thus, a general response to acidosis may be the induction of the integrated stress response, a cytoprotective pathway initiated in response to exposure to various environmental stimuli [39, 40], with ATF4 as central player [41].

We then considered all predicted upstream regulators belonging to the category "transcription regulators" and we filtered for those with adjusted p-values below 0.05 for each cell type. This resulted in a list of 32 predicted upstream regulators. However, these had Z-scores with non-coherent directionality and, in part, values well below 2 (S3 Table). Functional enrichment analysis by g:Profiler for this set of upstream regulators indicates an impact on processes related to cellular stress ("integrated stress response"), carcinogenesis and ageing (S4 Table), besides the expected RNA-polymerase II transcription process. In line with the IPA analysis above, the terms "TGF-beta signaling pathway" and "TP53 network" were also predicted (Table 1).

Applying the same process without restriction of the predicted upstream regulators to the category "transcription regulators", 83 upstream regulators (S3 Table) were identified. Functional enrichment analysis by g:Profiler with this set of regulators yielded similar results, i.e. impact on processes related to cell stress, carcinogenesis and ageing. In addition, an impact on ErbB signalling pathways is predicted that overlap with processes related to carcinogenesis.

**Table 1. Selected predicted upstream regulators for acidosis-induced gene expression changes (from Fig 6a and S3 Table).** IPA calculated the Z-scores (positive and negative Z-scores reflect predicted activation and inhibition, respectively) and p-values. "n/a" stands for undetermined Z-scores.

| | | | Heatmap in Fig 6a | | | | Further functional enrichment analysis | |
|---|---|---|---|---|---|---|---|---|
| **Upstream Regulators** | | | **ATF4** | **CEBPB** | **NFE2L2** | **NUPR1** | **TGFB1** | **TP53** |
| Human | HK2 | Z-score | n/a | -4.68 | n/a | 5.81 | 1.81 | 3.77 |
| | | -log(BHpvalue) | - | 18.50 | - | 14.15 | 9.53 | 18.61 |
| | CCD1092Sk | Z-score | -2.46 | -0.97 | 2.10 | -2.70 | -0.42 | 0.51 |
| | | -log(BHpvalue) | 5.47 | 1.67 | 2.92 | 5.27 | 21.18 | 10.39 |
| | HAoSMC | Z-score | -4.12 | 0.49 | 1.96 | 0.29 | 2.76 | 1.14 |
| | | -log(BHpvalue) | 18.65 | 4.33 | 4.44 | 8.03 | 25.60 | 15.48 |
| Rat | NRK-52E | Z-score | 4.07 | -3.54 | 3.93 | 3.74 | -4.62 | -0.83 |
| | | -log(BHpvalue) | 9.09 | 9.39 | 12.46 | 24.67 | 30.63 | 59.61 |
| | NRK-49F | Z-score | 2.64 | 3.90 | 3.49 | -2.78 | -2.69 | -4.86 |
| | | -log(BHpvalue) | 9.02 | 10.10 | 4.47 | 19.01 | 28.14 | 48.81 |

## Comparative downstream analysis of the transcriptional changes

While the "Upstream Regulator Analysis" predicted potential regulators responsible of the acidosis-regulated genes, the "Canonical Pathways" analysis predicted pathways putatively affected by the observed gene expression changes. An overview of the results is also available as heatmap (Fig 6b, detailed values available in S3 Table) and bubble charts (S2 Fig). No pathway was predicted as identically regulated in all cell types. However, a clear species-specific effect was visible, as pathways appeared putatively activated or inhibited in only one organism (Fig 6b), or even predicted in the opposite direction in the two organisms (Table 2). Furthermore, there was a predominance of metabolic pathways in human cells as compared to signalling pathways in rat cells (S2 Fig) that might result from a more robust wiring of metabolism in rats.

## Discussion

Extracellular pH is an important parameter with influence on cell function and fate. Microen-vironmental acidosis accompanies different pathological situations, including tumours, inflammation, hypoxia, ischemia and fibrosis. Research focussed mainly on acidification of the tumour micromilieu e and the possible consequences on proliferation, migration and drug resistance. Hereby, the impact of extracellular acidosis on the transcriptome of tumour cells has been investigated [42–44]. Much less is known regarding the impact of local acidosis on the transcriptome of non-tumour cells, which can be exposed to these conditions during inflammation, hypoxia, ischemia, fibrosis or metabolic derailment.

In the present study, we investigated the impact of metabolic acidosis on the transcriptome, within the cell type-individual resilience range, on five non-tumour cell types of human and rat origin. As our data show, there was no universal "acidosis-transcriptome", i.e. a strong uni-form pattern of transcriptional changes observed in all five cell types with their individual aci-dosis resilience range. We could identify only four genes whose expression was sensitive to acidosis in all cell types. Out of these four genes, only two responded concordantly.

The relevance of these two "acidosis" genes is most probably different. MYOM1 is a myo-cyte specific protein and could be involved in myocyte responses to prolonged local acidosis, e.g. during ischemia or inflammation. However, we could detect reasonable expression levels (> 5 FPM) only in CCD1092Sk cells. Therefore, MYOM1 cannot be considered a prominent acidosis response gene. By contrast, nuclear receptor coactivator 5 (NCOA5) is an ubiquitous protein and is expressed as well as downregulated in all five investigated cell types. NCOA5

**Table 2. Selected predicted canonical pathways for acidosis-induced gene expression changes (from Fig 6b).** IPA calculated the Z-scores (positive and negative Z-scores reflect predicted activation and inhibition, respectively) and p-values. "n/a" stands for undetermined Z-scores.

| Canonical pathway | | | Integrin signaling | Paxillin Signaling |
|---|---|---|---|---|
| Human | HK2 | Z-score | n/a | n/a |
| | | -log(BHpvalue) | - | - |
| | CCD1092Sk | Z-score | 2.00 | 2.11 |
| | | -log(BHpvalue) | 2.75 | 4.04 |
| | HAoSMC | Z-score | 2.40 | 1.27 |
| | | -log(BHpvalue) | 1.05 | 1.35 |
| Rat | NRK-52E | Z-score | -3.07 | -1.21 |
| | | -log(BHpvalue) | 6.08 | 4.13 |
| | NRK-49F | Z-score | -3.22 | -2.27 |
| | | -log(BHpvalue) | 3.75 | 3.51 |

acts as regulator of the transcriptional activity of e.g. the nuclear estrogen receptors alpha and beta, and the retinoic acid-related orphan receptor-alpha [30, 45–47]. Thus, NCOA5 could be a general transducer of acidic stress to the transcription regulation machinery. Reduced expression of NCOA5 has been associated with carcinogenesis, glucose intolerance (T2D), stem cell function as well as proinflammatory and profibrotic alterations of the micromilieu [30, 48, 49]. Thus, it is conceivable that acidosis-induced downregulation of NCOA5 contributes to the pathological alterations during local extracellular acidosis.

Because the number of affected genes in the five cell types did not correlate with the degree of acidosis, we exclude the possibility that the differences result from different pH-values (which were all at the lower edge of the resilience range). The number of affected genes in NRK-52E cells differs substantially from the one in NRK-49F cells, and the number in HAoSMC from the ones in HK-2 and CCD1092Sk cells. Rather, the results of our quantitative analysis support the hypothesis that there is no uniform acidosis-transcriptome but that the responses to acidotic stress are species- and cell type-specific, in quantitative and qualitative terms. This knowledge regarding species specificity is of relevance for translation efforts of data from animal studies to humans and vice versa. With respect to acidosis stress in non-tumour cells, transferability seems unfortunately to be limited.

The qualitative assessments of acidosis-induced transcriptome changes made by functional enrichment analysis (g:Profiler) and upstream analysis (IPA) confirm the strong impact of cell type and species on the acidosis responses. NUPR1, ATF4, NFE2L2 and CEBPB are part of the prominent potential acidosis-effect regulators we identified. They are all well-known transcriptional regulators that act in response to cell stress, mainly oxidative and endoplasmic reticulum stress, and regulate cell survival by a mechanism called integrated stress response [34–38]. Thus, a general response to acidosis may be the modulation of this integrated stress response, a cytoprotective pathway initiated in response to exposure to various environmental stimuli [39, 40], with ATF4 as central player [41]. The core event in this pathway is the phosphorylation of the eukaryotic translation initiation factor 2 alpha, eIF2$\alpha$, which can lead to a decrease in global protein synthesis and the induction of certain genes, including the transcription factor ATF4. The induced transcriptome optimizes the response to stress as a function of the cellular context as well as the nature and intensity of the stress. Furthermore, the TGF-beta signalling and TP53 networks are predicted for all cell types with high adjusted p-values, although Z-scores varied. By contrast, NFE2L2 was regulated significantly and concordant in four cells types. The NFE2L2 signaling network acts cytoprotectively in a variety of stress situations and prevents damage from enhanced reactive oxygen formation. For example, NFE2L2 prevents damage progression after ischemia/reperfusion due to the upregulation of genes that regulate redox balance and the supply of NADPH and other cellular fuels [50–53]. Furthermore, it has been shown to be acidosis-responsive [13]. NFE2L2 regulates genes that coordinate homeostatic processes to prevent tissue damage, including protective enzymes such as peroxidases, reductases and transferases [54]. Taken together, these analysis results indicate that there possibly exists a small general set of acidosis-responsive cellular networks.

Although we have performed a comprehensive and in-depth analysis in five cells types from human and rat, comprising epithelial and mesenchymal cells, it still represents a bioinformatic-derived, hypothesis-generating prediction. However, due to the large amount of data included, it is a prediction with high evidence and therefore represents a robust basis for the development of advanced working hypotheses. The next step will be the functional and biological validation of the acidosis-responsive cellular networks, in parallel to the biochemical assessment of NCOA5.

In summary, we provide strong evidences for a predominant cell type- and species-specific acidosis response. Nevertheless, we still identified the integrated stress response

(ISR), TGF-beta signalling, NFE2L2 and TP53 as four general acidosis-responsive cellular networks. In addition, the regulator of the transcriptional activity, NCOA5, is a promising candidate for a general acidosis-responsive gene. Future studies will have to extend our bio-informatics analysis to cell biological validation experiments, to test the refined working hypothesis.

## Supporting information

**S1 File. Comprehensive summary of the results obtained by Ingenuity Pathway Analysis (IPA) for all cell types.**
(PDF)

**S1 Fig. Graphical summaries of the results obtained by Ingenuity Pathway Analysis (IPA) for all cell types.**
(PDF)

**S2 Fig. Graphical presentation of the canonical pathway analysis for all cell types.**
(PDF)

**S1 Table. Master file of RNASeq data for all cell types.**
(XLSX)

**S2 Table. gProfiler_multiquery analysis of differentially expressed genes.**
(XLSX)

**S3 Table. Upstream regulator and canonical pathway analysis of differentially expressed genes by IPA.**
(XLSX)

**S4 Table. gProfiler_analysis predicted upstream regulators by IPA.**
(XLSX)

## Author Contributions

**Conceptualization:** Michael Gekle.

**Data curation:** Virginie Dubourg, Marie-Christin Schulz, Stefanie Ruhs, Michael Gekle.

**Formal analysis:** Virginie Dubourg, Marie-Christin Schulz, Philipp Terpe, Stefanie Ruhs.

**Funding acquisition:** Michael Gekle.

**Investigation:** Virginie Dubourg, Marie-Christin Schulz, Philipp Terpe, Stefanie Ruhs, Michael Kopf.

**Methodology:** Virginie Dubourg, Marie-Christin Schulz, Philipp Terpe, Stefanie Ruhs, Michael Kopf, Michael Gekle.

**Project administration:** Michael Gekle.

**Supervision:** Michael Gekle.

**Writing – original draft:** Virginie Dubourg, Michael Gekle.

**Writing – review & editing:** Virginie Dubourg, Marie-Christin Schulz, Philipp Terpe, Stefanie Ruhs, Michael Gekle.

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
