## [Decision Letter · Decision Letter 0]

2 Jun 2023

PONE-D-23-10137Hypothesis-generating analysis of the impact of non-damaging metabolic acidosis on the transcriptome of different cell types: Integrated stress response (ISR) modulation as general transcriptomic reaction to non-respiratory acidic stress?PLOS ONE

Dear Dr. Gekle,

Thank you for submitting your manuscript to PLOS ONE. After careful consideration, we feel that it has merit but does not fully meet PLOS ONE’s publication criteria as it currently stands. Therefore, we invite you to submit a revised version of the manuscript that addresses the points raised during the review process. Please submit your revised manuscript by Jul 17 2023 11:59PM. If you will need more time than this to complete your revisions, please reply to this message or contact the journal office at plosone@plos.org. Please include the following items when submitting your revised manuscript:A rebuttal letter that responds to each point raised by the academic editor and reviewer(s). You should upload this letter as a separate file labeled 'Response to Reviewers'.A marked-up copy of your manuscript that highlights changes made to the original version. You should upload this as a separate file labeled 'Revised Manuscript with Track Changes'.An unmarked version of your revised paper without tracked changes. You should upload this as a separate file labeled 'Manuscript'.If applicable, we recommend that you deposit your laboratory protocols in protocols.io to enhance the reproducibility of your results. Protocols.io assigns your protocol its own identifier (DOI) so that it can be cited independently in the future. For instructions see: https://journals.plos.org/plosone/s/submission-guidelines#loc-laboratory-protocols. Additionally, PLOS ONE offers an option for publishing peer-reviewed Lab Protocol articles, which describe protocols hosted on protocols.io. Read more information on sharing protocols at https://plos.org/protocols?utm_medium=editorial-email&utm_source=authorletters&utm_campaign=protocols.

We look forward to receiving your revised manuscript.

Kind regards,

Kanhaiya Singh, Ph.D

Academic Editor

PLOS ONE

https://journals.lww.com/jasn/pages/default.aspx

https://docksci.com/acidic-environment-activates-inflammatory-programs-in-fibroblasts-via-a-camp-map_5a7bbb2cd64ab22d4c439d9f.html

https://pubs.acs.org/doi/10.1021/acs.jproteome.5b00503

https://www.mdpi.com/2076-3921/12/2/412

https://academic.oup.com/hmg/article/26/R2/R91/3978054

In your revision ensure you cite all your sources (including your own works), and quote or rephrase any duplicated text outside the methods section. Further consideration is dependent on these concerns being addressed.

Additional Editor Comments:

Please respond to the comment of the reviewer to validate the finding at RT-PCR level.

Reviewers' comments:

Reviewer's Responses to Questions

**Comments to the Author**

1. Is the manuscript technically sound, and do the data support the conclusions?

Reviewer #1: Yes

Reviewer #2: Yes

2. Has the statistical analysis been performed appropriately and rigorously? 

Reviewer #1: Yes

Reviewer #2: Yes

3. Have the authors made all data underlying the findings in their manuscript fully available?

Reviewer #1: Yes

Reviewer #2: Yes

4. Is the manuscript presented in an intelligible fashion and written in standard English?

Reviewer #1: Yes

Reviewer #2: Yes

5. Review Comments to the Author

Reviewer #1: Extracellular pH is an important parameter influencing cell function and fate. This article exhibits the transcriptional impact of metabolic acidosis on five non-tumour cell types of human and rat origin by RNA-Seq and bioinformatics analysis. Dual differential-expression analysis (DESeq and edgeR) and, after appropriate homology mapping, Gene Ontology enrichment analysis (g:Profiler), Ingenuity Pathway Analysis (IPA), as well as functional enrichment analysis were used for predicted upstream regulators, and the regulator of the transcriptional activity NCOA5 as the only general acidosis-responsive gene.

While there were several spell mistakes below:

1. In page9, paragraph 3, line 7, research should be changed as researches.

2. In page11, paragraph 1, line 3, CO2 should be changed as CO2.

3. In page22, Figure 4, line 1,shows should be changed as show.

4. I think RT-PCR experiments should be used to verify the RNS-seq data.

5. The English writing should be largely improved.

Reviewer #2: The manuscript entitled "Hypothesis-generating analysis of the impact of non-damaging metabolic acidosis on

the transcriptome of different cell types: Integrated stress response (ISR) modulation as general transcriptomic reaction to non-respiratory acidic stress?" is written in well defined way and can be accepted for the publications.

6. PLOS authors have the option to publish the peer review history of their article (what does this mean?). If published, this will include your full peer review and any attached files.

Reviewer #1: No

Reviewer #2: No

---

## [Author Response · Author response to Decision Letter 0]

15 Jun 2023

Dear Editor,

herewith we submit our revised manuscript entitled “Hypothesis-generating analysis of the impact of non-damaging metabolic acidosis on the transcriptome of different cell types: Integrated stress response (ISR) modulation as general transcriptomic reaction to non-respiratory acidic stress?”

We thank the reviewers for their comments and their recommendation to publish our manuscript. All comments raised by the reviewers and editorial office were addressed in the revised manuscript version, as detailed below.

Datasets generated and/or analyzed during the current study are available in the gene expression omni-bus database. Human data: GSE220788 (token: glkxqmgclpofpiv). Rat data: GSE220789 (token: obcrcmkirbmbbgd).

All authors agree with the submission. The work has not been published or submitted for publication elsewhere. There is no conflict of interest. 

Detailed responses to the issues raised:

We revised the manuscript according to the style templates above.

We went again thoroughly through the entire manuscript and checked it for grammar and spelling errors. As we are experienced in the use of the English language for scientific publications (and also spend time in Australia and the US), we think we are able to evaluate a scientific text concern-ing appropriate English writing.

A copy of our manuscript showing all changes by either using track changes is provided as a sup-porting information file.

A clean copy of the edited manuscript has been uploaded as the new manuscript file.

3. We noticed you have some minor occurrence of overlapping text with the following previous publica-tion(s), which needs to be addressed:

We compared the text of the mentioned publications (3 of the mentioned publications are from our group) with the text of our manuscript, sentence by sentence, but could not detect major overlaps. Nevertheless, we modified few sentences in the introduction that showed some similarities due to the basic and general nature of the issues described. 

https://journals.lww.com/jasn/pages/default.aspx

Although not specified we assume that it refers to following publication: Wesson DE, Buysse JM, Bushinsky DA (2020) Mechanisms of Metabolic Acidosis-Induced Kidney Injury in Chronic Kidney Disease. J Am Soc Nephrol 31: 469-482. The authors of this review describe, as we do in the introduction, mechanisms and cellular consequences of local acidosis. Because we were aware that similar issues are described for which the variability of wording is limited, we cited this re-view several times in the introduction (Ref. 3). We modified few sentences in the introduction that showed some similarities due to the basic and general nature of the issues described.

https://docksci.com/acidic-environment-activates-inflammatory-programs-in-fibroblasts-via-a-camp-map_5a7bbb2cd64ab22d4c439d9f.html

This is a publication of our group that also deals with the impact of acidosis. Therefore, a cer-tain overlap in wording, especially in the introduction and the method section, may occur. Howev-er, we could not detect overlaps outside the method section. Ref 16. A. Riemann, … M. Gekle. Biochimica et Biophysica Acta 1853 (2015) 299–307. 

https://pubs.acs.org/doi/10.1021/acs.jproteome.5b00503

This is a publication of our group that also deals with the impact of acidosis. Therefore, a cer-tain overlap in wording, especially in the introduction especially in the introduction and the method section, may occur. However, we could not detect relevant overlaps outside the method section. Nevertheless, we modified one sentence. A. Ihling, … M. Gekle. J. Proteome Res. 2015, 14, 9, 3996–4004

https://www.mdpi.com/2076-3921/12/2/412

This is a publication of our group that also deals with the impact of acidosis. Therefore, a cer-tain overlap in wording, especially in the introduction especially in the introduction and the method section, may occur. However, we could not detect overlaps outside the method section. Ref. 13. MC Schulz, … M. Gekle. Antioxidants 2023, 12(2), 412

https://academic.oup.com/hmg/article/26/R2/R91/3978054

This is a completely unrelated publication that is unknown to us. This study deals with a neuro-pathological topic, amyotrophic lateral sclerosis, which is not related to our area of expertise. In their manuscript the authors discuss the general cell biological mechanism of “integrated stress response” as we do and cite a publication that we also cite (Pakos-Zebrucka K, Koryga I, Mnich K, Ljujic M, Samali A, Gorman AM (2016) The integrated stress response. EMBO rep 17: 1374-1395.). Integrated stress response is a clearly defined cellular mechanism, the characterization of which does not allow room for diverse explanations. Therefore, you may find overlapping wording whenever integrated stress response is described or discussed. However, we could not identify overlapping wording in terms of partial or even entire phrases.

We checked our revision carefully concerning the appropriate citation of all sources.

4. We note that the grant information you provided in the ‘Funding Information’ and ‘Financial Disclo-sure’ sections do not match.

When you resubmit, please ensure that you provide the correct grant numbers for the awards you re-ceived for your study in the ‘Funding Information’ section.

We added the correct information: This project was supported funded by the Deutsche For-schungsgemeinschaft (DFG GE 905/19). The funders had no role in study design, data collection and analysis, decision to publish, or preparation of the manuscript.

5. We note that you have indicated that data from this study are available upon request. 

This must be a misunderstanding, since we declared “Yes - all data are fully available without re-striction”. This is documented in the PONE-S-23-12894 file. Datasets generated and/or analyzed during the current study are available in the gene expression omnibus database. Human data: GSE220788 (token: glkxqmgclpofpiv). Rat data: GSE220789 (token: obcrcmkirbmbbgd).

6. Please include captions for your Supporting Information files at the end of your manuscript, and up-date any in-text citations to match accordingly. Please see our Supporting Information guidelines for more information: http://journals.plos.org/plosone/s/supporting-information.

Done.

Done.

Additional Editor Comments:

Please respond to the comment of the reviewer to validate the finding at RT-PCR level.

We performed a comprehensive bioinformatic analysis on RNA sequencing results comprising a large number of genes from five different cell types, to identify patterns and generate new hypoth-esis. This analysis includes bioinformatically derived upstream predictions concerning the activity of signaling modules as well as downstream predictions concerning cellular functions. These pa-rameters cannot be validated by RT-PCR, but have to be investigated at the protein and/or func-tional level. These analyses are the focus of future projects and are beyond the scope of the pre-sent study. Furthermore, high quality RNA sequencing with the appropriate number of biological replicates followed by state–of-the-art differential expression analysis, including extensive quality control and normalization, leads to more reliable and valid results compared to RT-PCR with one or two housekeepers. We are convinced that the next step must be the determination of relevant alterations at the protein level (changes in expression or in modification, like phosphorylation) and/or functional cell assays. Currently we are setting up the methodology for these work packag-es.

Reviewers' comments:

1. In page9, paragraph 3, line 7, research should be changed as researches.

The word “research” does not appear on page 9. On page 3 there is the phrase “In the past, re-search focused mainly on tumor microenvironment acidosis as key parameter for cell fate and disease progression.” This phrase is grammatically correct.

2. In page11, paragraph 1, line 3, CO2 should be changed as CO2.

On page 5 CO2 was changed to CO2.

3. In page22, Figure 4, line 1,shows should be changed as show.

Shows was changed to show.

4. I think RT-PCR experiments should be used to verify the RNS-seq data.

We performed a comprehensive bioinformatical analysis on RNA sequencing results comprising a large number of genes from five different cell types, to identify patterns and generate new hypoth-esis. This analysis includes bioinformatically derived upstream predictions concerning the activity of signaling modules as well as downstream predictions concerning cellular functions. These pa-rameters cannot be validated by RT-PCR, but have to be investigated at the protein and/or func-tional level. These analyses are the focus of future project and are beyond the scope of the present study. Furthermore, high quality RNA sequencing with the appropriate number of biological repli-cates followed by state-of-the art differential expression analysis, including extensive quality con-trol and normalization, leads to more reliable and valid results compared to RT-PCR with one or two housekeepers. We are convinced that the next step must be the determination of relevant al-terations at the protein level (changes in expression or in modification, like phosphorylation) and/or functional cell assays. Currently we are setting up the methodology for these work packag-es.

5. The English writing should be largely improved.

Both reviewers answered question 4 above (“Is the manuscript presented in an intelligible fashion and written in standard English?”) with yes. Thus, the comment raised here is contradictory. As we are experienced in the use of the English language for scientific publications (and also spend time in Australia and the US), we think we are able to evaluate a scientific text concerning appro-priate English writing. Nevertheless, we went again thoroughly through the entire manuscript and checked it for grammar and spelling errors.

---

## [Decision Letter · Decision Letter 1]

8 Aug 2023

Hypothesis-generating analysis of the impact of non-damaging metabolic acidosis on the transcriptome of different cell types: Integrated stress response (ISR) modulation as general transcriptomic reaction to non-respiratory acidic stress?

PONE-D-23-10137R1

Dear Dr. Gekle,

We’re pleased to inform you that your manuscript has been judged scientifically suitable for publication and will be formally accepted for publication once it meets all outstanding technical requirements.

Kind regards,

Kanhaiya Singh, Ph.D

Academic Editor

PLOS ONE

Additional Editor Comments (optional):

Reviewers' comments:

Reviewer's Responses to Questions

**Comments to the Author**

1. If the authors have adequately addressed your comments raised in a previous round of review and you feel that this manuscript is now acceptable for publication, you may indicate that here to bypass the “Comments to the Author” section, enter your conflict of interest statement in the “Confidential to Editor” section, and submit your "Accept" recommendation.

Reviewer #1: All comments have been addressed

2. Is the manuscript technically sound, and do the data support the conclusions?

Reviewer #1: Yes

3. Has the statistical analysis been performed appropriately and rigorously? 

Reviewer #1: Yes

4. Have the authors made all data underlying the findings in their manuscript fully available?

Reviewer #1: Yes

5. Is the manuscript presented in an intelligible fashion and written in standard English?

Reviewer #1: Yes

6. Review Comments to the Author

Reviewer #1: (No Response)

7. PLOS authors have the option to publish the peer review history of their article (what does this mean?). If published, this will include your full peer review and any attached files.

Reviewer #1: No

---

## [Editor Report · Acceptance letter]

17 Aug 2023

PONE-D-23-10137R1 

Hypothesis-generating analysis of the impact of non-damaging metabolic acidosis on the transcriptome of different cell types: Integrated stress response (ISR) modulation as general transcriptomic reaction to non-respiratory acidic stress? 

Dear Dr. Gekle:

I'm pleased to inform you that your manuscript has been deemed suitable for publication in PLOS ONE. Congratulations! Your manuscript is now with our production department. 

Kind regards, 

on behalf of

Dr. Kanhaiya Singh 

Academic Editor

PLOS ONE